# Selective Layer Tuning and Performance Study of Pre-Trained Models Using Genetic Algorithm

Jae-Cheol Jeong [1,†] , Gwang-Hyun Yu [2,†] , Min-Gyu Song [1] , Dang Thanh Vu [2] , Le Hoang Anh [2] , Young-Ae Jung [3] , Yoon-A Choi [4] , Tai-Won Um [5,*] and Jin-Young Kim [2,*]

1    Department of Biomedical Engineering, Chonnam National University Hospital, Gwangju 61469, Korea
2    Department of ICT Convergence System Engineering, Chonnam National University, Gwangju 61186, Korea
3    Division of Information Technology Education, Sunmoon University, Asan 31460, Korea
4    Korea Electric Power Research Institute (KEPRI), Daejeon 34056, Korea
5    Graduate School of Data Science, Chonnam National University, Gwangju 61186, Korea
*    Correspondence: stwum@jnu.ac.kr (T.-W.U.); beyondi@jnu.ac.kr (J.-Y.K.)
†    These authors contributed equally to this work.

**Abstract:** Utilizing pre-trained models involves fully or partially using pre-trained parameters as initialization. In general, configuring a pre-trained model demands practitioners' knowledge about problems or an exhaustive trial–error experiment according to a given task. In this paper, we propose tuning trainable layers using a genetic algorithm on a pre-trained model that is fine-tuned on single-channel image datasets for a classification task. The single-channel dataset comprises images from grayscale and preprocessed audio signals transformed into a log-Mel spectrogram. Four deep-learning models used in the experimental evaluation employed the pre-trained model with the ImageNet dataset. The proposed genetic algorithm was applied to find the highest fitness for every generation to determine the selective layer tuning of the pre-trained models. Compared to the conventional fine-tuning method and random layer search, our proposed selective layer search with a genetic algorithm achieves higher accuracy, on average, by 9.7% and 1.88% (MNIST-Fashion), 1.31% and 1.14% (UrbanSound8k), and 2.2% and 0.29% (HospitalAlarmSound), respectively. In addition, our searching method can naturally be applied to various datasets of the same task without prior knowledge about the dataset of interest.

**Keywords:** deep learning; selective layer tuning; genetic algorithm; pre-trained model

## 1. Introduction

Recent deep-learning-based projects have employed pre-trained models according to the data domain and project environment. A pre-trained model attains high performance through a long training time and high computing power using various training techniques with big datasets. In particular, in computer vision, deep-learning models trained on the ImageNet dataset [1] demonstrated good results in classification and detection. Moreover, in natural language processing, models such as transformer [2], generative pre-trained transformer (GPT) [3], and Bidirectional Encoder Representations from Transformers (BERT) [4] trained using the Wikipedia dataset [5] showed good performance in the field of translation. By using a pre-trained model trained with big data in this manner, the time for designing, learning, and verifying deep-learning models can be saved when solving similar problems. However, as actual projects have problems and datasets from various fields, the pre-trained model should be fine-tuned. Generally, in fine-tuning a pre-trained model, there is a considerable difference in the performance and learning time of the pre-trained model according to the user's knowledge and experience.

The urge to use pre-trained models has increased as deep-learning models have proved their effectiveness in many industrial applications. However, the way to effectively apply a pre-trained model has rarely been studied directly, and there has been less previous

evidence for employing a pre-trained model across different fields. Instead, other studies focus on proposing a task-specific module [6–8] that only benefits a few domains or developing a search algorithm to find decent deep neural architectures from scratch [9], which is non-practical. To fill in the literature gap, this study proposes a search procedure that assists pre-trained models when fine-tuning to solve a new problem. Our method deploys a genetic algorithm to mutually seek effective layers that perform well on a specific dataset for the classification task. The experimental results on various datasets and state-of-the-art deep neural architectures show the robustness of our proposed method both in accuracy and complexity measured by the number of model parameters.

The contributions of this work are summarized as follows:

- We present a search procedure with the genetic algorithm to search for selective layers of a pre-trained deep-learning model. It yielded significant speed advantages when starting with a pre-trained model rather than other search-from-scratch approaches.
- We introduce HospitalAlarmSound, a dataset about alarm sounds recorded from medical appliances at the Hospital of Chonnam National University. The dataset is carefully designed for the classification task with 8 classes and 569 records in total.

We conducted experiments on two public datasets (MNIST-Fashion, UrbanSound8k) and our alarm sound dataset, in which a sample was later converted into a Mel-spectrogram image. Additionally, to show that our method is not limited by model type, we have run experiments with four well-known models whose pre-trained versions have been extensively used by deep-learning practitioners.

This paper is structured as follows: The introduction and research trend of the pre-trained model and genetic algorithm are presented in Section 2. The overview of the research is presented in Section 3. Further, the experiments and results of the single-channel-dataset-based pre-trained model are described in Section 4. Finally, Section 5 presents the conclusion. All codes used in the experiments have been shared at https://github.com/GWANGHYUNYU/Audio/tree/main/GA, accessed on 20 June 2022.

## 2. Related Works

### 2.1. Fine-Tuning of Pre-Trained Models

In general, training a deep-learning model requires a large amount of data; however, obtaining big data is difficult in reality. In addition, when deep-learning models are trained from scratch, various trial-and-error processes are required to determine hyperparameters. To overcome this drawback, a pre-trained model based on large-scale training data can be utilized.

The methods of using the pre-trained model can be broadly categorized into three types, as shown in Figure 1: using only the structure of the model, learning only some layers and fixing the remaining layers, and using the model as a feature extractor. Based on this categorization, there are four ways to use a pre-trained model according to the characteristics of the researcher's dataset:

1. The size of a user's data is small, while the similarity to the pre-trained model data is high: Because the data similarity is very high, there is no need to retrain the pre-trained model or modify the classifier to fit the task (modify the dense layer and softmax layer). This is a method of using a pre-trained model as a feature extractor.
2. The size of the user's data is small, and the similarity to the pre-trained model data is low: Freeze the initial layers of the pre-trained model, retrain only the remaining layers, and modify the classifier to fit the task. The advantages of the pre-trained model can be maximized by using the low-level features of the pre-trained model as they are and changing only the high-level features to fit its data.
3. The user's data are large but less similar to the pre-trained model data: Because of the large size of the user's data, the pre-trained model structure is imported, the classifier is modified to fit the task, and then it is trained from scratch. Because the data similarity is very low, the weights and biases of the pre-trained model adversely affect its performance.

4.  The user's data are large and highly similar to the pre-trained model data: The structure, weight, and bias of the pre-trained model are used as they are, the classifier is modified to fit the task, and it is trained again with its data. In an ideal situation, the pre-trained model can be used most effectively.

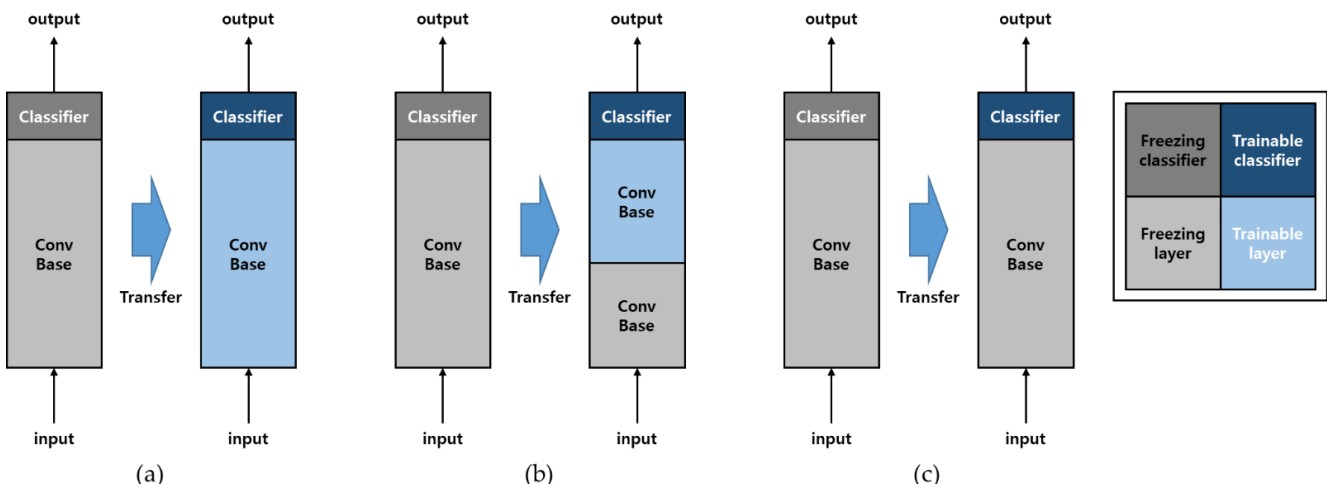

**Figure 1.** Methods of fine-tuning a pre-trained model: (**a**) using only the structure; (**b**) training only some layers and freezing the rest; (**c**) using the pre-trained model as a feature extractor.

### 2.2. Research Related to Fine-Tuning

Fine-tuning research using a pre-trained model is applied to various tasks in several datasets. There have been several reports in this research field. For example, in a garbage separation and collection classification study, Inception-V4, DenseNet, and MobileNetV1 were fine-tuned based on the TrashNet dataset [10]. In an art classification study, the fully connected layer and classifier part of CaffeNet were fine-tuned based on WikiArt, WGA, and TICC datasets [11]. Next, in a study on plant disease identification, Inception-V4, VGG, ResNet, and DenseNet were fine-tuned based on the Plant Village Dataset [12]. Furthermore, BERT was fine-tuned in a medical relation extraction study based on corpus datasets such as VCDR, TCM, and i2b2 temporal relation [13]. In a breast cancer classification study, ResNet, DenseNet, MobileNetV2, and ShuffleNetV2 were fine-tuned based on the breast thermal image dataset [14]. Further, in a study on American Sign Language translation, AlexNet and GoogleNet were fine-tuned based on the ASL dataset [15]. In a heart sound classification study, pre-trained audio neural networks were fine-tuned based on a heart sound database [16]. Moreover, in a pedestrian detection study, using a mobile phone with a 4-MP camera, Faster R-CNN + InceptionV2, SSD + InceptionV2, and SSD + MobileNetV2 were fine-tuned based on street video data [17]. In addition, cutterhead torque prediction [18] and atrial fibrillation detection [19] automatically extract informative deep features from the source and target domains and then feed them into the feature predictor for final results.

Thus, the fine-tuning of a pre-trained model has been applied to diverse datasets, from trash and plant datasets similar to ImageNet datasets to completely different art, sign language, and thermal images. In addition, there are datasets with different characteristics from images such as corpus, sound, and video, where the task has been solved by fine-tuning the pre-trained model suitable for the dataset. Finally, deep features were extracted from datasets in the source and target domains and used in the predictor.

### 2.3. Genetic Algorithms

A genetic algorithm represents a probabilistic search method that imitates natural selection and the genetic rule, which are the evolutionary processes in living things. The genetic algorithm begins with a population of individuals representing the latent solution of a problem. The population maintains a certain number of objects in each generation.

Further, each generation evaluates the fitness of each individual and probabilistically selects the objects to survive in the next generation accordingly. Some of the selected individuals randomly mate to produce offspring. In this case, the genes of the parents are inherited by the offspring through crossover, and mutations may occur. Assuming that the offspring inherit good genetic traits from their parents, the potential solutions of the next generation are, on average, better than those of the previous generation. This evolutionary process is repeated until the termination condition is satisfied.

*2.4. Genetic Algorithm-Based Fine-Tuning Research*

The research on fine-tuning a pre-trained deep-learning model based on genetic algorithms has been applied to limited datasets as a study to enhance performance. In an effective deep neural network (DNN) structure discovery study, the best structure was selected by applying a genetic algorithm to the DNN model based on the AURORA2 dataset [20]. In a DNN study for forecasting in an outpatient department, the best structure was selected by applying a genetic algorithm to the DNN model based on the OPD dataset [21]. Further, in a deep-learning model study for image classification, the best layer was selected by applying a genetic algorithm using InceptionV3 and ResNet models based on the Disaster, Network Camera 10K, CIFAR10, and MNIST-Fashion datasets [22]. In addition, an ensemble convolutional neural network (CNN) model study for crop pest classification determined the weighted average ensemble by applying genetic algorithms to VGG, ResNet, InceptionV3, Xception, MobileNet, and SqeezeNet based on the Insect dataset [23]. Moreover, in a study on transfer learning layer selection, the best layer was selected by applying the genetic algorithm to the InceptionV3 model based on the CIFAR100 dataset [24]. Next, in a study on an appropriate dataset-based fine-tuning method, the best fine-tuning was obtained by applying the genetic algorithms in the multilayer perceptron and long short-term memory models based on the CIFAR10, CIFAR100, IMDB, and SST datasets [25]. As described above, genetic algorithms are applied to improve fine-tuning performance, model structure, and ensemble optimization of pre-trained models based on each dataset.

**3. Method**

Even though a pre-trained model is a quick response to solve machine learning problems, the model is often limited to a single data type or task. For example, the pre-trained ResNet on the ImageNet dataset for classification is unlikely to be transferable to solve NLP tasks such as language modeling or machine translation, but it probably performs well on other image-related tasks such as object detection or segmentation. For this reason, we limit this study's domain to the one-channel image classification task, in which we focus on dealing with one-channel image type and using pre-trained models over the ImageNet.

*3.1. Dataset*

The single-channel dataset used in the proposed selective layer tuning of pre-trained deep-learning models by a genetic algorithm comprised the MNIST-Fashion dataset [26], UrbanSound8K dataset [27], and the HospitalAlarmSound dataset that we collected ourselves. The MNIST-Fashion dataset contains grayscale images of size $28 \times 28 \times 1$ that are resized to $32 \times 32 \times 1$. Because the UrbanSound8K and HospitalAlarmSound datasets contain audio signals, we changed them to a one-channel image format through feature extraction, as shown in Figure 2.

$$\text{dB} = 20log_{10}|X_k|, \tag{1}$$

$$\text{Mel}(f) = 2595\text{log}\left(1 + \frac{f}{700}\right). \tag{2}$$

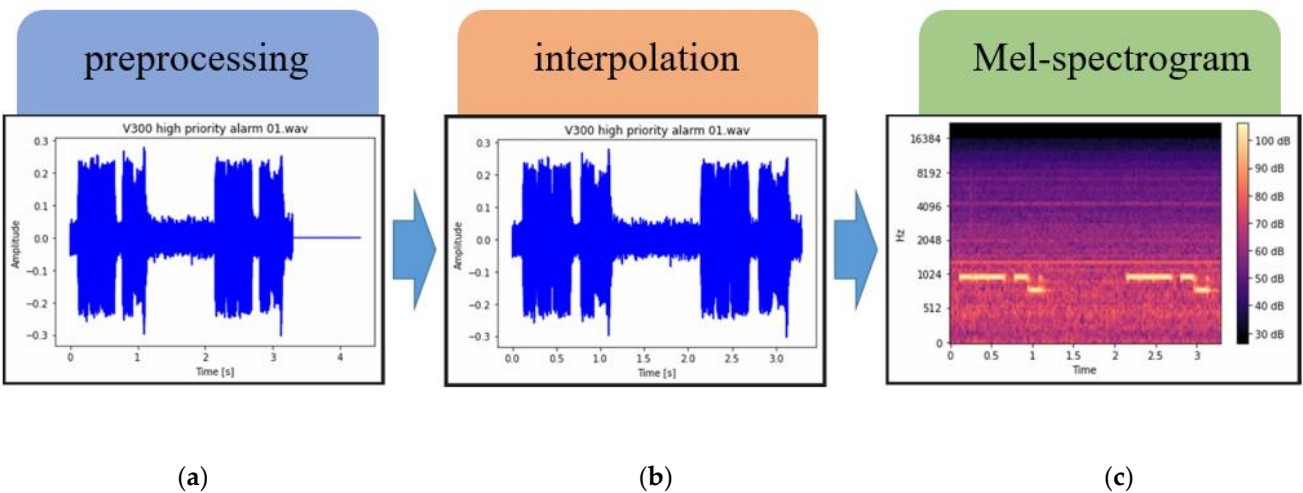

**Figure 2.** Preparation for audio data. The process involves three steps, preprocessing (**a**), interpolation (**b**), and Mel-Spectrogram transformation (**c**).

Audio signal preprocessing involves cutting a meaningful section of an audio file in advance and adjusting the entire audio file to the same length as that of the cut section. A spectrum is obtained by applying Fourier transform to the preprocessed audio file and sliding by as much as a window. As shown in Equation (1), the log-spectrum is obtained by applying the decibel (dB) unit of the log scale to the spectral value. If this log-spectrum is stacked side by side as much as sliding the window, a spectrogram having the log-spectrum value as much as the window on the time axis can be obtained. If the Mel-scale, which reflects human auditory characteristics, is applied to this spectrogram as in Equation (2), a Mel spectrogram is obtained. As shown in Figure 2c, the *x*-axis denotes time, the *y*-axis denotes frequency, and the decibel (color bar) representing the intensity of the frequency is plotted on the *z*-axis.

The self-collected HospitalAlarmSound dataset records the alarm sound from patient-monitoring devices installed in hospital rooms, as shown in Figure 3. This is a recording of eight alarm sounds from five patient-monitoring devices. During data collection, each piece of patient-monitoring equipment was manually operated in a quiet space, and recording was performed for a time when the pattern of the alarm could be checked. We record the sound signal with a default frequency of 44 kHz, 25 min and 46 s long. The average length of samples is 2.72 s; the longest recording lasts 3.6 s, and the shortest lasts 0.4 s. Table 1 gives details of the classes of the entire dataset and the number of data values per class.

The purpose of introducing the HospitalAlarmSound is twofold. Firstly, a sample was recorded in a near-laboratory environment with static settings such as a tranquil room, a single recorder, and common sources of the speaker, meaning the number of noise samples is near zero. Additionally, a speaker was independently recorded, so there was no mix of multiple speakers. Secondly, the source was domain-specific when only the alarm sound from medical appliances was considered, which is both a pro and con of the dataset. While the dataset is easy to interpret and the inner class variance is small, it is insufficient for learning a general model.

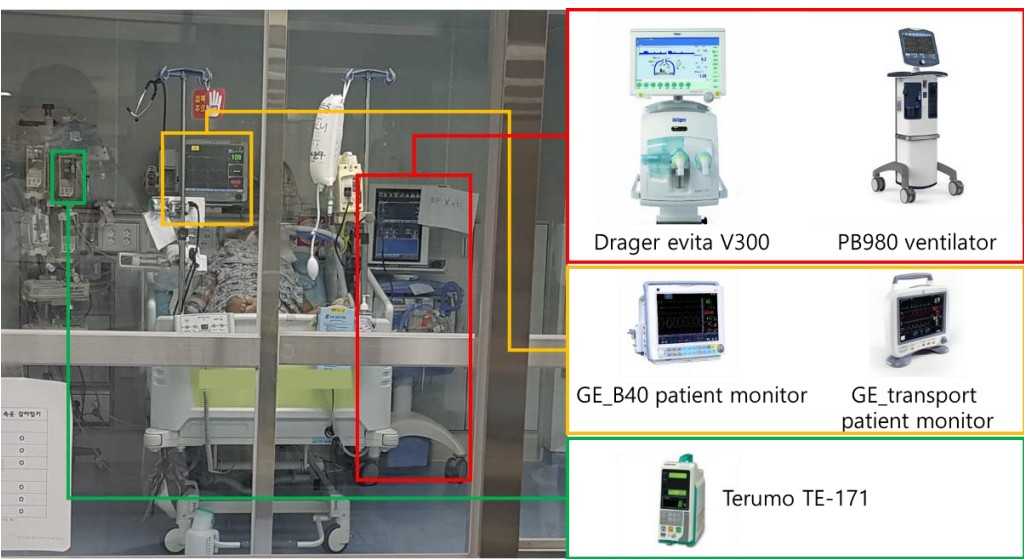

**Figure 3.** Collection setup for HospitalAlarmSound dataset.

**Table 1.** Summary of datasets.

| Dataset | Class | Number | Dataset | Class | Number | Dataset | Class | Number |
|---|---|---|---|---|---|---|---|---|
| MNIST-fashion | T-shirt/top | 7000 | Urban Sound8K | air_conditioner | 1000 | Hospital Alarm Sound | Dager evita V300 | 70 |
| | Trouser | 7000 | | car_horn | 429 | | GE_B40-high | 72 |
| | Pullover | 7000 | | children_playing | 1000 | | GE_B40-medium | 69 |
| | Dress | 7000 | | dog_bark | 1000 | | GE_transport-advisory | 71 |
| | Coat | 7000 | | drilling | 1000 | | GE_transport-crisis | 71 |
| | Sandal | 7000 | | engine_idling | 1000 | | GE_transport-warning | 75 |
| | Shirt | 7000 | | gun_shot | 374 | | PB980 | 65 |
| | Sneaker | 7000 | | jackhammer | 1000 | | TE-171 | 76 |
| | Bag | 7000 | | siren | 929 | | | |
| | Ankle boot | 7000 | | street_music | 1000 | | - | |
| Total | | 70,000 | Total | | 8732 | Total | | 569 |

### 3.2. Pre-Trained Deep-Learning Models

The pre-trained deep-learning model used for selective layer tuning based on a genetic algorithm is a CNN structure that is in the spotlight as a deep-learning-based image-classification model. CNNs have been continuously studied since 1998 with LeNet [28], achieving high accuracy in a number-recognition dataset based on the convolutional layer, max pooling, and fully connected layer. In 2012, AlexNet [29] produced innovative results in the 1000 classification competition released by ImageNet and demonstrated that convolutional neural networks performed well in image classification. In 2014, VGGNet [30] and GoogleNet [31], which are close to the 5% error of the human image classification test, appeared. They overcame the limit of eight layers that could be stacked uniquely, demonstrating high recognition rates with 19 and 22 deep networks. In 2015, ResNet [32] implemented 152 deep networks through its own identity short connection idea, exceed-

ing the recognition rate of human classification. Subsequently, various models such as Inception-ResNetV4 and DenseNet have been reducing errors every year based on ResNet's short connection idea. Since then, NASNet [33], which proposed a CNN with a new structure by applying Auto-ML to input data based on reinforcement learning, was researched. Further, the MobileNet-series [34] and ShuffleNet-series [35] models reduced the number of parameters and improved the processing speed required for training. EfficientNet [36], which proposes an optimal model structure for data by improving the input image size and hyperparameters, has recently been widely used. In the present work, VGGNet, ResNet, MobileNetV1, and EfficientNet were selected as pre-trained models, and weights and biases were trained from 1000 class ImageNet datasets.

### 3.3. Selective Layer Tuning by Genetic Algorithm

Figure 4 presents an overview of the selective layer tuning by the genetic algorithm proposed in this paper. All layers of the pre-trained model trained with the ImageNet dataset were selected as the optimal trainable and freezing layers based on the genetic algorithm for the given dataset. The detailed algorithm for selective layer tuning by a genetic algorithm is presented in Table 2.

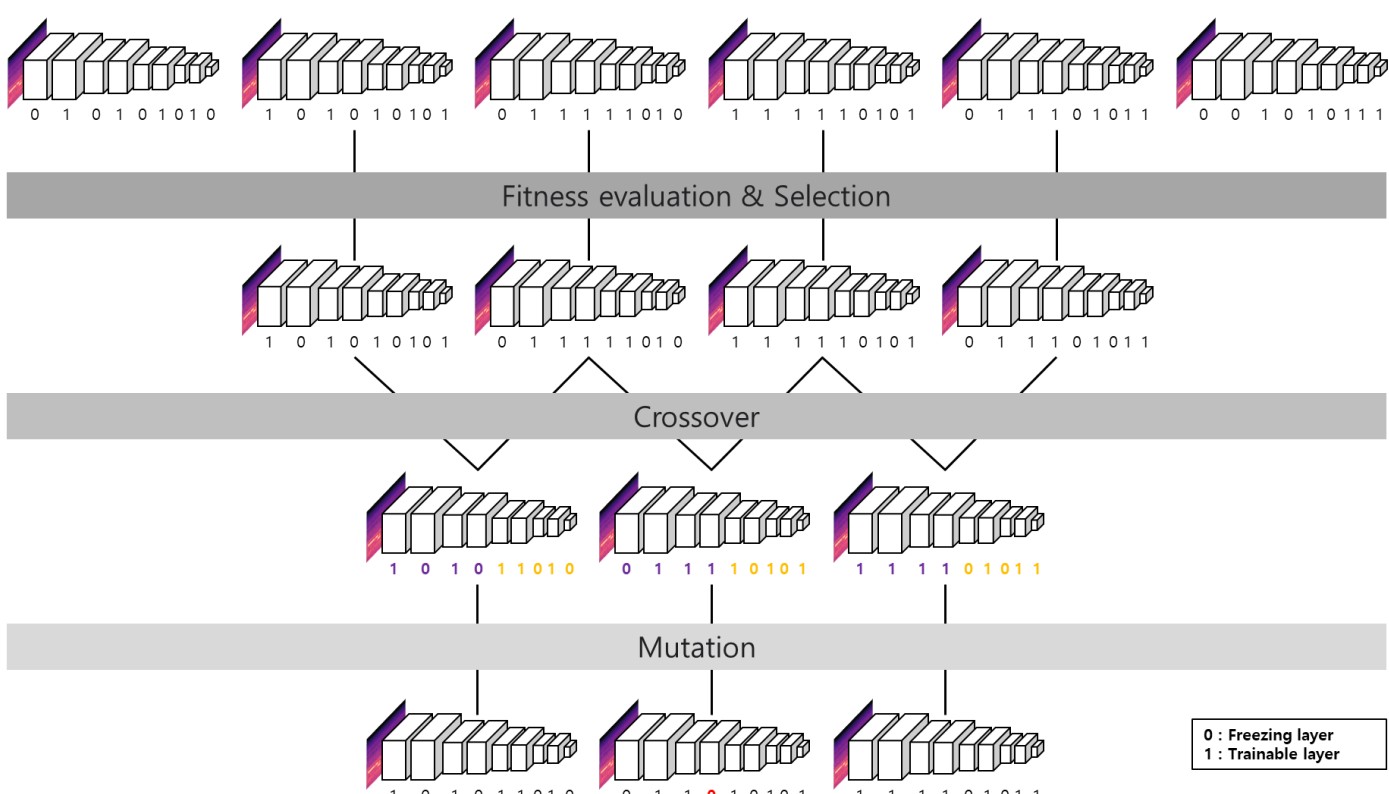

**Figure 4.** Overview of selective layer tuning based on genetic algorithm in a pre-trained deep-learning model.

**Table 2.** Layer selection for transfer learning using genetic algorithms.

| | |
|---|---|
| 1: | Generate: $n$ initial genomes |
| 2: | while Generation < Final Generation: |
| 3: | Train: $n$ models corresponding to $n$ genomes |
| 4: | Select: $n/2$ genomes based on top $n/2$ fitness score |
| 5: | Crossover1: $n/4$ child genomes based on a regular crossover |
| 6: | Crossover2: $n/4$ child genomes based on a random crossover |
| 7: | Mutation: $n/2$ genomes and $n/2$ child genomes |
| 8: | Align: $n/2$ genomes and $n/2$ child genomes to the next generation |
| 9: | Generation $+ = 1$ |
| 10: | end While: |
| 11: | Return: Array of the selected layers |

1. Genome: The gene is a pre-trained model trained with the ImageNet dataset that allows all layers to be selected as a trainable layer and a freezing layer.
2. Initial generation: In the first generation, the entire layer for each genome is randomly selected as the trainable layer and freezing layer.
3. Fitness evaluation: Short training is performed on a given dataset based on randomly selected trainable and freezing layers, and validation accuracy is obtained for each epoch. The highest fitness score is ranked by setting a validation accuracy as a fitness indicator.
4. Selection: Choose from the top dominant genomes selected through fitness evaluation.
5. Crossover: Select and cross over the selected dominant genomes. Two main crossover methods are used, as shown in Figure 5. The crossover of the half-mixing method according to the high fitness score is shown in Figure 5a. The selected dominant genomes are crossed over randomly to make the crossover more effective, as shown in Figure 5b.

   Crossover is a genetic operator that combines the genetic information of two parents to generate new offspring. This study uses regular and random crossover simultaneously to enhance the variety of the solution space through exploration, which could help avoid being corrupted early at local optima. Both regular crossover and random crossover are designed based on the one-point crossover, while the difference between them is the portion the offspring inherit from the parent generation. In the regular crossover, we generate the offspring by taking half of the genome code from two selected genomes. On the other hand, the offspring is constructed by an arbitrary crossover point in the random crossover when the random range is dependent on the size of the dominant genomes.
6. Mutation: In all genomes of the current generation, a layer is randomly selected with a 4% probability and reversed.
7. Next generation: Half of all genomes are selected as dominant genomes, and the dominant genomes are converted into child genomes using two crossover methods. All selected genomes undergo mutation. Finally, the total number of dominant and child genomes will be the same as the initial number of populations in the next generation.
8. Iteration: Selection, crossover, mutation, and next generation are repeated until the target is achieved.

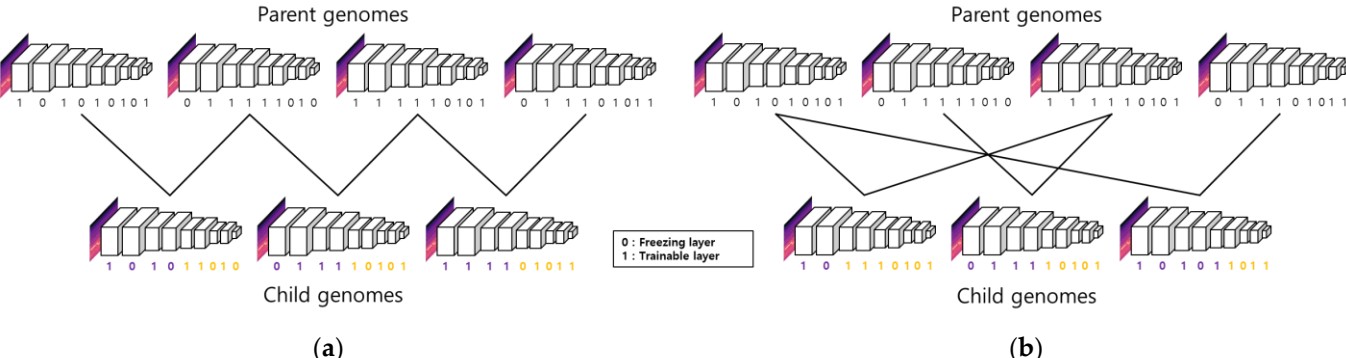

**Figure 5.** Two crossover methods: (**a**) general crossover method; (**b**) random crossover method.

## 4. Experiments and Results

In the proposed selective layer tuning by the genetic algorithm, we investigated whether the selection of layers to be trained or frozen in fine-tuning the layers of pre-trained models was ideal for a given dataset. Accordingly, experiments were performed with four pre-trained models on three single-channel datasets using a genetic algorithm.

### 4.1. Experimental Environment

The hardware specification of the deep-learning server on which the experiment was performed was as follows: Intel(R) Xeon(R) W-2123 CPU, NVIDIA RXT 3060 VRAM 12 GB, and 32 GB RAM. The pre-trained model and genetic algorithm were configured using the Python-based TensorFlow 2.7 Framework. The UrbanSound8K dataset was used by resampling 44.1 kHz to 22.05 kHz, and it converted the length of all data items to 4 s, zero-padding the data under 4 s. After preprocessing, feature extraction was performed to obtain the Mel spectrogram and finally generate a $128 \times 173 \times 1$ image. It was divided into 7859 training data and 873 test data. Finally, the HospitalAlarmSound dataset used the same 44.1 kHz sampling rate as the collection environment and converted the length of all data items to 3.6 s, zero-padding the data under 3.6 s. After preprocessing, feature extraction was performed for obtaining the Mel spectrogram and generating an image of size $128 \times 311 \times 1$. It was divided into 456 training data and 113 test data. Additionally, the data were augmented by adding noise and shift to the zero-padding dataset, and the Mel spectrogram was applied. The augmented dataset is divided into 3756 training data and 2503 testing data.

### 4.2. Experimental Results

VGG16, ResNet50, MobileNetV1, and EfficientNetB0 were applied to each dataset using the selective layer tuning of the pre-trained model by the genetic algorithm described in Section 3.3. The genome generated in the corresponding generation was a randomly selected layer-tuned model. Upon comparing the fitness scores of the genomes, we found an excellent selective-layer-tuning model. The fitness score is the validation accuracy of the model for the experiment. Therefore, the hyperparameters of genetic algorithms are divided into two types: epoch, batch size, and learning rate needed to train deep-learning models, and the number of populations and number of iterations for the generation of the genetic algorithm. The default setting of a hyperparameter was as follows: epoch = 5, batch size = 256, learning rate = 0.0001, number of populations = 30, and number of iterations = 20. In the case of insufficient memory when experimenting with a large pre-trained model, the batch size and number of populations were reduced by half.

If the genetic algorithm was executed using the set hyperparameter, each generation's array of selective layers for the dominant genomes was returned. For the performance test, the pre-trained model was fine-tuned by selecting the genome with the highest validation accuracy among the returned genomes. Fine-tuning settings—epoch = 50, batch size = 512,

learning rate = 0.0001—were the same as the hyperparameter settings of the genetic algorithm.

For the performance comparison experiment, among the fine-tuning methods, training only the classifier without learning all the layers of the pre-trained model, and training both the pre-trained model and some layers and classifiers of the pre-trained model were implemented. In addition, a method of randomly training pre-trained models and classifiers was employed. In particular, we group the cases, including fine-tuning with fully freezing, half freezing, and zero freezing, as the heuristic method. In the fully freezing case, all model's weights except for the classifier are set as non-trainable, while in the half freezing case, half of the shallow layers are set as non-trainable, and in the case of zero freezing, all model's weights are set as trainable. On the other hand, the random method occurs when the freezing layers are randomly picked regarding the model architecture. We suggest the heuristic and random methods according to the common habit of practitioners when deploying a pre-trained model. Notably, the random method can be interpreted as the initial generation of the genetic algorithm. For each dataset and pre-trained model, we compare the classification performance of the above-mentioned methods with our best architecture sought by the genetic algorithm.

As presented in Tables 3 and 4, in the case of the MNIST-Fashion and UrbanSound8K datasets, experimental results show that tuning the selective layer using the genetic algorithm is more effective than fine-tuning the pre-trained model or random selection layer tuning. In addition, all genetic algorithm-based selective layer tuning methods can reduce the number of trainable parameters required for fine-tuning an existing pre-trained model by up to one-sixth. In the case of HospitalAlarmSound dataset, the difference according to data class is clearly distinguished and the number of data is small. Therefore, experimenting only with the classifier while leaving the pre-trained model layer achieved 100% accuracy. Additionally, the experimental results of the augmented HospitalAlarmSound dataset are shown in Table 5. The selective layer tuning based on the genetic algorithm is better than fine-tuning the pre-trained model or random selection layer tuning. It can be seen that the number of learnable parameters was less than that of the general fine-tuning methods.

For each dataset, we calculate the average performance gap over models according to the following formula:

$$\overline{acc} = \frac{\sum_j^m \left( \sum_i^n \frac{rg_j - r_{ji}}{n} \right)}{m} \tag{3}$$

where $rg$ is the test accuracy of the genetic algorithm and $r$ is the test accuracy of method $i \in \{zero - freezing, \ fully - freezing, \ half - freezing\} \cup \{ random \}$ from the model $j \in \{EfficientNetB0, \ ResNet50, \ MobileNetV1, \ VGG16\}$. Using Equation (3), the average performance gaps between the heuristic and random methods and our method are 9.7% and 1.88% (MNIST-Fashion), 1.31% and 1.14% (UrbanSound8k), 2.2% and 0.29% (HospitalAlarmSound), respectively.

**Table 3.** Experiment results of MNIST-Fashion dataset-based selective layer tuning by genetic algorithm.

| Model | Attributes | Heuristic Method | | | Random Method | Genetic Algorithm |
|---|---|---|---|---|---|---|
| EfficientNetB0 | number of trainable layers | 0 | 237 | 137 | 121 | 128 |
| | test accuracy | 0.4940 | 0.8495 | 0.7895 | 0.8181 | 0.8738 |
| | number of parameters | 12,816 | 4,020,364 | 3,490,948 | 2,051,830 | 1,767,472 |
| ResNet50 | number of trainable layers | 0 | 175 | 100 | 89 | 81 |
| | test accuracy | 0.7850 | 0.9022 | 0.9032 | 0.9030 | 0.9144 |
| | number of parameters | 20,496 | 23,555,088 | 19,473,424 | 12,627,728 | 10,876,944 |
| MobileNetV1 | number of trainable layers | 0 | 86 | 46 | 39 | 43 |
| | test accuracy | 0.5033 | 0.9297 | 0.9035 | 0.9300 | 0.9319 |
| | number of parameters | 10,256 | 3,217,232 | 2,942,480 | 1,037,200 | 1,476,624 |
| VGG16 | number of trainable layers | 0 | 19 | 9 | 11 | 6 |
| | test accuracy | 0.8512 | 0.9401 | 0.9324 | 0.9348 | 0.9441 |
| | number of parameters | 5136 | 14,719,824 | 12,984,336 | 9,668,112 | 6,091,216 |

**Table 4.** Experiment results of UrbanSound8K dataset-based selective layer tuning by genetic algorithm.

| Model | Attributes | Heuristic Method | | | Random Method | Genetic Algorithm |
|---|---|---|---|---|---|---|
| EfficientNetB0 | number of trainable layers | 0 | 237 | 137 | 122 | 129 |
| | test accuracy | 0.9599 | 0.9920 | 0.9908 | 0.9840 | 0.9931 |
| | number of parameters | 307,216 | 4,314,764 | 3,785,348 | 2,082,326 | 2,509,558 |
| ResNet50 | number of trainable layers | 0 | 175 | 100 | 91 | 96 |
| | test accuracy | 0.9851 | 0.9851 | 0.9782 | 0.9828 | 0.9920 |
| | number of parameters | 491,536 | 24,026,128 | 19,944,464 | 10,552,400 | 13,297,104 |
| MobileNetV1 | number of trainable layers | 0 | 86 | 46 | 53 | 40 |
| | test accuracy | 0.9737 | 0.9840 | 0.9828 | 0.9759 | 0.9931 |
| | number of parameters | 204,816 | 3,411,792 | 3,137,040 | 3,049,264 | 522,192 |
| VGG16 | number of trainable layers | 0 | 19 | 9 | 8 | 5 |
| | test accuracy | 0.9817 | 0.9588 | 0.9748 | 0.9794 | 0.9897 |
| | number of parameters | 102,416 | 14,817,104 | 13,081,616 | 5,119,504 | 5,008,336 |

**Table 5.** Experiment results of HospitalAlarmSound dataset-based selective layer tuning by genetic algorithm.

| Model | Attributes | Heuristic Method | | | Random Method | Genetic Algorithm |
|---|---|---|---|---|---|---|
| EfficientNetB0 | number of trainable layers | 0 | 237 | 137 | 125 | 114 |
| | test accuracy | 0.9872 | 0.9952 | 0.9880 | 0.9940 | 0.9956 |
| | number of parameters | 512,016 | 4,519,564 | 3,990,148 | 2,583,768 | 2,182,418 |
| ResNet50 | number of trainable layers | 0 | 175 | 100 | 90 | 80 |
| | test accuracy | 0.9400 | 0.9736 | 0.9952 | 0.9956 | 0.9980 |
| | number of parameters | 819,216 | 24,353,808 | 20,272,144 | 9,197,456 | 4,526,544 |
| MobileNetV1 | number of trainable layers | 0 | 86 | 46 | 41 | 46 |
| | test accuracy | 0.9053 | 0.9968 | 0.9924 | 0.9936 | 0.9996 |
| | number of parameters | 368,656 | 3,575,632 | 3,300,880 | 2,351,024 | 2,864,560 |
| VGG16 | number of trainable layers | 0 | 19 | 9 | 9 | 6 |
| | test accuracy | 0.9568 | 0.9840 | 0.9824 | 0.9928 | 0.9944 |
| | number of parameters | 184,336 | 14,899,024 | 13,163,536 | 7,302,480 | 5,606,608 |

Table 6 presents the results of an experiment comparing the MNIST-Fashion Dataset with other genetic algorithm-based model-selection methods. The proposed method obtained better results in precision and recall than a method of creating a new model through complex mixing of the best layers from N models based on a genetic algorithm. For the best result of fine-tuning the classifier while leaving the pre-trained model layer as it is, none of the other fine-tuning, random selective layer-tuning, and genetic algorithm-based selective layer-tuning methods gave good results.

**Table 6.** Comparison of experiment results of MNIST-Fashion dataset.

| Method | Precision | Recall |
|---|---|---|
| Tian, H. [16] | 0.9289 | 0.9292 |
| Proposed Method (MobileNetV1) | 0.9331 | 0.9310 |
| Proposed Method (VGG16) | 0.9451 | 0.9401 |

Figure 6a,b illustrate the highest validation accuracy among the selective layer tuning results from generation 1 to generation 20 using a genetic algorithm. An analysis of the 1st–20th generations of the genetic algorithm reveals that it evolves to find the best performance with each generation. The experiment confirmed that the evolution proceeds rapidly from the 1st to the 15th generation and gradually converges from the 15th generation onwards. Following the experiment's result, rather than infinitely training the genetic algorithm, configuring up to 20 generations as the maximum number of iterations is more effective. Finally, even for the same pre-trained model, selective layer tuning can be adopted differently for a given dataset, as shown in Figures 7 and 8. Figures 7 and 8 show the results of the performance of the genetic algorithm-based selective layer tuning on the MNIST-Fashion and UrbanSound8K datasets using VGG16 as a pre-trained model.

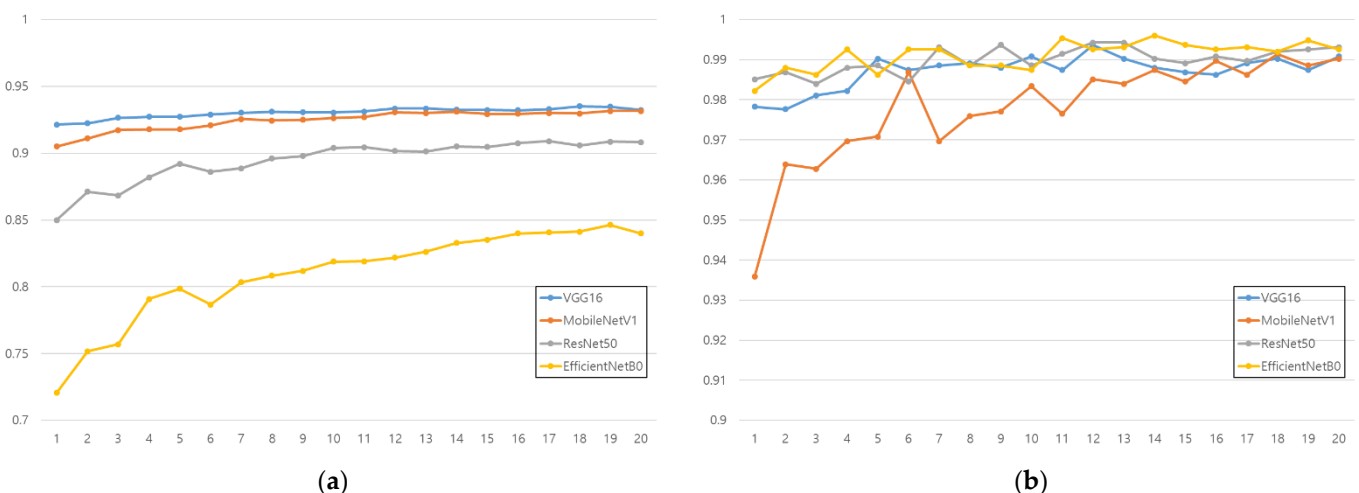

(**a**)        (**b**)

**Figure 6.** Accuracy results of two pre-trained models by each generation: (**a**) MNIST-Fashion dataset-based genetic algorithm; (**b**) UrbanSound8K dataset-based genetic algorithm.

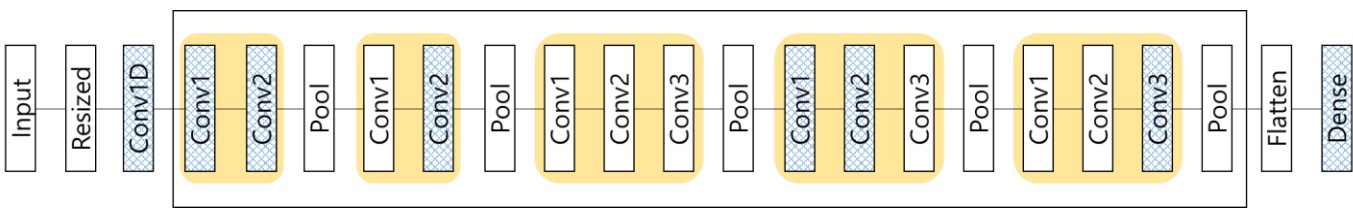

**Figure 7.** MNIST-Fashion dataset-based selective layer tuning by VGG16.

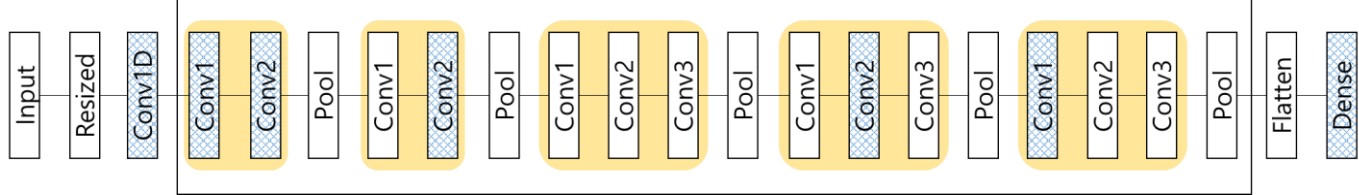

**Figure 8.** UrbanSound8K dataset-based selective layer tuning by VGG16.

The best results for the same pre-trained model are obtained by selecting and training different layers for the given dataset. Figures 7–9 indicate that each dataset selects the most effective convolutional layer instead of the general VGG16 block unit.

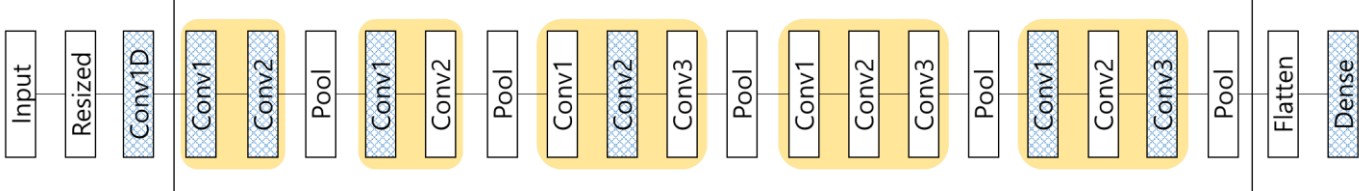

**Figure 9.** HospitalAlarmSound dataset-based selective layer tuning by VGG16.

## 5. Conclusions

Recent deep-learning-based projects have frequently fine-tuned pre-trained models to solve their own problems. Instead of fatigue trial-and-error episodes, we argue that strategically searching for an appropriate pre-trained architecture is feasible. In this work, a CNN-based pre-trained model showing good performance in image classification was trained by selective layer tuning based on a genetic algorithm.

In the experiment on grayscale and Mel-spectrogram images, selective layer tuning based on a genetic algorithm showed better performance than the heuristic tuning or fine-tuning of the pre-trained model, or tuning the selective layer at random. In addition, selective layer tuning based on a genetic algorithm can obtain high performance while minimizing the number of trainable parameters required for fine-tuning a pre-trained model. Thus, given a pre-trained model suitable for various tasks, a high-performance tuned pre-trained model can be easily and conveniently obtained according to the data. Compared to the heuristic and random methods, our proposed selective layer search achieves higher accuracy by, on average, 9.7% and 1.88% (Fashion-Mnist), 1.31% and 1.14% (UrbanSound8k), 2.2% and 0.29% (HospitalAlarmSound), respectively.

This research has several limitations. We narrow our study to a single task as image classification with only one-channel image datatype. In addition, our neural architecture search is based on the genetic algorithm, so the proposed algorithm naturally inherits its drawbacks, such as exponential complexity and local optima. We are certain that the presented method can be generalized to other tasks and data in future research.

**Author Contributions:** Conceptualization, J.-C.J. and J.-Y.K.; methodology, G.-H.Y., Y.-A.C. and J.-Y.K.; software, L.H.A.; validation, G.-H.Y., D.T.V. and L.H.A.; formal analysis, D.T.V.; investigation, Y.-A.C. and T.-W.U.; resources, M.-G.S.; data curation, Y.-A.C. and M.-G.S.; writing—original draft preparation, G.-H.Y.; writing—review and editing, J.-C.J.; visualization, L.H.A.; supervision, T.-W.U. and J.-Y.K.; project administration, Y.-A.J., T.-W.U. and J.-Y.K.; funding acquisition, Y.-A.J. and J.-Y.K. All authors have read and agreed to the published version of the manuscript.

**Funding:** This work was supported by the Korea Electric Power Research Institute (KEPRI) grant funded by the Korea Electric Power Corporation (KEPCO) (No. R20IA02). This work was also supported by an Institute of Information & Communications Technology Planning & Evaluation (IITP) grant funded by the Korean government (MSIT) (No. 2021-0-02068, Artificial Intelligence Innovation Hub).

**Institutional Review Board Statement:** Not applicable.

**Informed Consent Statement:** Not applicable.

**Data Availability Statement:** MNIST-Fashion dataset can be found at https://github.com/zalando research/fashion-mnist (accessed on 20 June 2022), UrbanSound8K dataset can be found at https://urbansounddataset.weebly.com/urbansound8k.html (accessed on 20 June 2022), and HospitalAlarm-Sound dataset can be obtained at https://github.com/GWANGHYUNYU/Audio/tree/main/GA.

**Conflicts of Interest:** The authors declare no conflict of interest.

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
