# Peer review of "Selective Layer Tuning and Performance Study of Pre-Trained Models Using Genetic Algorithm"

_electronics, doi:10.3390/electronics11192985_

Round 1
Reviewer 1 Report
This paper on GA based selective layer tuning of DL networks is interesting and relevant for practitioners in the field. The introduction, related work, experimental results, and conclusion sections are well written and the authors' motivation and intentions clear.
The methodology section is uneven in its descriptive level, e.g. the introduction of the 3 datasets intersperses the description of sample preprocessing with the class breakdown for each dataset. It's unlikely that a reader unfamiliar with a Mel spectrogram would grasp the process based on the description given, so why not further abbreviate the description and instead provide a fuller description of the hospital alarm dataset? Why not add something about the motivation for including it?
The GA section could also use a bit more clarity as to the crossover operator. Is there a single split point or does the split vary? I think this is what is intended by the random crossover option, but it's not clear from the writing. How does the random crossover work?
There should be some description of the heuristic and random training methods that appear in the results tables. While I can infer what the heuristics are and can imagine what a random method might be, these should be clearly defined for the reader. These definitions could appear either in an additional methodology section or could be added to the experimental results introduction.
Finally, the hospital alarm dataset appears to be dismissed in the results section. Even if the results aren't that interesting compared to the other datasets something more should be included, otherwise why bother introducing them in the first place?
Author Response
- The methodology section is uneven in its descriptive level, e.g. the introduction of the 3 datasets intersperses the description of sample preprocessing with the class breakdown for each dataset. It's unlikely that a reader unfamiliar with a Mel spectrogram would grasp the process based on the description given, so why not further abbreviate the description and instead provide a fuller description of the hospital alarm dataset? Why not add something about the motivation for including it?
Thank you for the comments from the reviewer. As transforming from 1D signal data into 2D image-based Mel spectrogram is a main preprocessing for not only our "HospitalAlarmSound" dataset but also the UrbanSound8K dataset, the authors believe that it is important to explain this step clearly. The information here is provided enough; thus the authors concern that further abbreviation will make readers mislead the details. Besides, the authors also agree with the reviewer that information about the dataset should be clarified more. The modifications can be tracked at lines 179-185, 216-219, 221-228.
- The GA section could also use a bit more clarity as to the crossover operator. Is there a single split point or does the split vary? I think this is what is intended by the random crossover option, but it's not clear from the writing. How does the random crossover work?
Thank you for the question from the reviewer. Both regular and random crossover is designed based on one-point crossover, while the difference is the portion that the offspring inherit from the parent generation. In the regular crossover, we generate the offspring by taking haft of the genome code from two selected genomes. On the other hand, the offspring is constructed by an arbitrary crossover point in the random crossover when the random range is dependent on the size of the selected genomes. The authors have added the explanations into the GA section, the modifications can be tracked at line 274-288
- There should be some description of the heuristic and random training methods that appear in the results tables. While I can infer what the heuristics are and can imagine what a random method might be, these should be clearly defined for the reader. These definitions could appear either in an additional methodology section or could be added to the experimental results introduction.
Thank you for the notice from the reviewer. The authors have added descriptions of models used for performance comparison. The modifications can be tracked at line 342-352
- Finally, the hospital alarm dataset appears to be dismissed in the results section. Even if the results aren't that interesting compared to the other datasets something more should be included, otherwise why bother introducing them in the first place?
Thank you for the comments from the reviewer. The authors agree that it was our mistake when did not add a clear quantitative result for the HospitalSoundAlarm dataset. The authors have revised the dataset(in details at line 315-318) and added its experimental results to Table 5. in the manuscript. Furthermore, we have also clearly claimed our performance using quantitative results, which have been provided at the abstract and conclusion section.

Reviewer 2 Report
1. The title of the manuscript needs to be revised. Any type of single channel data?
2. In the introduction, the author needs to clearly express the contributions and innovations of the paper. At present, the contribution and innovation of the paper are not clear.
3. In Section 2.2, some recent papars are missing. For example,
[a] A novel Domain Adaptive Residual Network for automatic Atrial Fibrillation Detection. Knowledge-Based System, 2020, 203:106122.
[b] An accurate and adaptative cutterhead torque prediction method for shield tunneling machines via adaptative residual long-short term memory network. Mechanical Systems and Signal Processing, https://doi.org/10.1016/j.ymssp.2021.108312.
4. Figure 1 and Figure 2 can be further beautified.
5. Are there quantitative results or conclusions in the conclusion section?
Author Response
- The title of the manuscript needs to be revised. Any type of single channel data?
Thank you for the suggestion from the reviewer. The authors determined that the paper name should be revised as "Selective layer tuning and performance study of pre-trained models using genetic algorithm" to highlight the contribution of the paper about "Selective layer tunning with genetic algorithm”. The single channel (alarm sound) dataset is a minor contribution to this paper.
- In the introduction, the author needs to clearly express the contributions and innovations of the paper. At present, the contribution and innovation of the paper are not clear.
Thank you for the suggestion from the reviewer. The authors have added statements verifying our paper's contributions and innovations. The modifications can be tracked at 49-75.
- In Section 2.2, some recent papers are missing. For example
Thank you for the suggestion, we have edited section 2.2 with your recommended citations. The modifications can be tracked at lines 134-137, 142-143.
- Figure 1 and Figure 2 can be further beautified.
Thank you for your comment, the authors have modified Figure 1 and Figure 2 more beautiful.
- Are there quantitative results or conclusions in the conclusion section?
Thank you for the notice from the reviewer, the authors have added quantitative results in the experiment section, as well as the abstract and conclusion section. The modifications can be tracked at lines 372-380, 421-430.

Round 2
Reviewer 2 Report
Congratulations! The revised manuscript has been improved.